# Rural-urban disparities in the nutritional status of younger adolescents in Tanzania

**Lorraine S. Cordeiro**[1]*, **Nicholas P. Otis**[1], **Lindiwe Sibeko**[1], **Jerusha Nelson-Peterman**[2]

**1** Department of Nutrition, School of Public Health and Health Sciences, University of Massachusetts Amherst, Amherst, MA, United States of America, **2** Department of Food and Nutrition, Framingham State University, Framingham, MA, United States of America

☯ These authors contributed equally to this work.

\* lcordeir@umass.edu

**Data Availability Statement:** All urban data files are available from the Harvard University Repository at https://doi.org/10.7910/DVN/9STGWE. All relevant rural data are within the manuscript and the full dataset is available at the

## Abstract

Research on geographic differences in health focuses largely on children less than five years; little is known about adolescents—and even less regarding younger adolescents—a vulnerable group at a critical stage of the life course. Africa's rapid population growth and urbanization rates, coupled with stagnant rates of undernutrition, further indicate the need for country-specific data on rural-urban health disparities to inform development policies. This study examined rural-urban disparities in body mass index-for-age-and-sex (BAZ) and height-for-age-and-sex z-scores (HAZ) among younger adolescents in Tanzania. Participants were randomly selected adolescents aged 10–14 years ($N = 1,125$) residing in Kilosa (rural) and Moshi (urban) districts of Tanzania. Individual and household-level data were collected using surveys and anthropometric data was collected on all adolescents. Age, sex, household living conditions, and assets were self-reported. BAZ and HAZ were calculated using the WHO reference guide. The prevalence of undernutrition was 10.9% among rural and 5.1% among urban adolescents ($p<0.001$). Similarly, stunting prevalence was greater in rural (64.5%) than urban (3.1%) adolescents ($p<0.001$). After adjusting for covariates, rural residence was significantly and inversely associated with BAZ ($B = -0.29$, 95% CI: -0.52, -0.70, $p = 0.01$), as well as with HAZ ($B = -1.79$, 95% CI: -2.03, -1.54, $p<0.001$). Self-identified males had lower BAZ ($B = -0.23$, 95% CI: -0.34, -0.11, $p<0.001$) and HAZ ($B = -0.22$, 95% CI: -0.35, -0.09, $p = 0.001$) than self-identified female adolescents. Rural-urban disparities in nutritional status were significant and gendered. Findings confirm place of residence as a key determinant of BAZ and HAZ among younger adolescents in Tanzania. Targeted gender-sensitive interventions are needed to limit growth faltering and improve health outcomes in rural settings.

## Introduction

Undernutrition is the leading cause of morbidity and mortality worldwide, contributing to an estimated 3.1 million deaths per year [1]. Existing literature across Sub-Saharan Africa (SSA)

ICPSR Data Sharing for Demographic Research repository (https://www.icpsr.umich.edu/web/pages/) by searching for project ID: DSDR-154081. Surveys for both datasets are included as Supporting Information files.

**Funding:** LSC - UNICEF/Tanzania Felton Earls and Mary Carlson - original data funded by U.S. National Institute of Mental Health (R01 MH66801) NPO - University of Massachusetts Amherst.

**Competing interests:** The authors have declared that no competing interests exist.

affirms an urban health advantage in relation to stunting [2–8], underweight [2,7], and body mass index (BMI) [6–8], suggesting that place of residence is an important social determinant of health outcomes. Most of the literature on geographic differences in health focuses on children under five years; little is known about adolescents, a vulnerable group at a critical stage of the life course. Emerging research on points of intervention to protect adolescent health, particularly during early adolescence, can present new policy and programmatic opportunities to promote child survival and improve adult health outcomes [1,9].

Adolescence, defined as 10–24 years of age [10], is second to infancy in linear growth velocity. This period of accelerated physical, hormonal, and cognitive development is accompanied by increased nutritional demand, which places adolescents in resource-constrained areas at greater risk for undernutrition. While research reveals the prevalence of undernutrition among adolescents in SSA ranges between 7% and 27% [3,6–8,11,12], there is a paucity of data examining adolescent health by rural versus urban residence. The few studies in this region lack consistency in findings and do not specifically address younger adolescents. While most of these studies confirmed an urban advantage in nutritional status, this study comprehensively examines age, sex, and place of residence as a determinant of adolescent nutritional health. Furthermore, this study adds to the limited knowledge of these intersections and presents disaggregated data on rural and urban adolescents in Tanzania, which aids the understanding of health risks affecting those transitioning from childhood into adulthood.

In Ethiopia, Berheto et al. (2015) found significantly lower mean BMI-for-age-and-sex z-scores (BAZ), lower mean height-for-age-and-sex z-scores (HAZ), and higher rates of stunting among rural versus urban adolescent girls [6]. Similarly, Hadley et al. (2011) reported lower weight-for-age-and-sex z-scores, BMI, HAZ, and BAZ among Ethiopian rural compared to urban adolescents after controlling for income, age, sex, and workload [7]. In Nigeria, Omigbodun et al. (2010) observed that rural adolescents had lower BAZ and were significantly more likely to be stunted and underweight than their urban peers [8]. Irenso et al. (2020) noted significantly lower HAZ and higher rates of stunting among rural compared to urban Ethiopian adolescents [3]. Lastly, in a small sample in Cameroon, Dapi et al. (2010) reported no differences in stunting among rural and urban adolescents [13].

Adolescents remain one of the most neglected, difficult to measure, and hard-to-reach populations. With rapid urban growth in SSA, a focus on adolescent health across geographic location is ever more vital to inform policy. Determinants of rural-urban health disparities can inform programmatic options to mitigate health consequences associated with migration and rapid urbanization, especially given that the nutritional status of the lowest quartiles of poor urban youth are on par with or worse than rural inhabitants [14,15].

Younger adolescents are often excluded from programs and policies that target children or young adults [16]. Few studies have examined place of residence as a determinant of nutritional health among younger adolescents and there are limited studies available to draw conclusions about urban-rural disparities in adolescent nutritional status across SSA. This study aimed to investigate rural/urban differences in BAZ and HAZ among adolescents, aged 10–14 years, living in Kilosa (rural) and Moshi (urban) districts of Tanzania. In addition to having one of the largest rural-urban gaps in malnutrition in SSA [17], 44% of Tanzania's population is under the age of 15 years and 70% of citizens live in rural areas [18]. Our study presents disaggregated data on adolescents from Tanzania, which is needed to inform health priorities and intervention points for adolescent health.

## Methods

### Study population

This cross-sectional comparative study merged and collectively analyzed two independent samples of adolescents (10–14 years) from Tanzania. In 2004, Cordeiro's study (Kilosa/rural) was conducted in rural Tanzania [11] and another study by Carlson and Earls (Moshi/urban) was conducted in an urban setting [19]. At the time of this study, Kilosa consisted of 161 registered rural villages [20] across 37 wards, and had a total population of 488,191. Moshi Urban District, the capital of the Kilimanjaro Region, comprised 15 wards totaling 143,799 people in 2002 [20].

The rural dataset was derived from a two-stage sampling plan for selection of villages and adolescents aged 10–19 years from Kilosa District, as detailed by Cordeiro et al. (2012) [11]. The Moshi/urban dataset was derived from a two-stage sampling plan for a cluster randomized control trial conducted in Moshi Urban District among adolescents aged 10–14 and their households [19]. From the combined sample, 1,237 were younger adolescents, aged 10–14 years. After eliminating 112 cases due to missing or incomplete data, the total analytical sample for this study was 1,125 adolescents aged 10–14 years.

### Data collection

The survey tool was developed in English based on an extensive literature review, and translated into Swahili by language experts in Tanzania, and then back-translated into English. The questionnaires were pre-tested among a sample of non-participants in Kilosa District and Moshi prior to using it with study participants. Trained interviewers administered the structured, pre-tested questionnaires in Swahili. Content validity was conducted, ensuring that terms and meanings were understood by participants in the pretest and that terms were accurately translated from English to Swahili. Construct validity required setting a priori hypotheses of associations and analyzing the pilot data to see if these hypotheses signaled validity. For quality assurance, piloting the survey tool among adolescents and their parents/guardian in both Kilosa and Moshi resulted in no major changes in the content and construct of the survey. Surveys in both English and Swahili are included as supporting information.

Both studies used 2002 Tanzanian census data to determine household composition, and all household demographics were verified during home visits. Surveys were administered in Swahili by trained interviewers who collected demographic, dietary, health, work status, developmental, and educational data. Adolescents and their caregivers were interviewed at the adolescent's residence or schools. Under the supervision of the research directors, height and weight were measured twice for each adolescent and averaged. Height was measured to the nearest 0.1cm with a standard stadiometer and weight was measured to the nearest 100g using calibrated UNICEF electronic scales. In the urban study, a senior research assistant repeated the physical exam measures on every fifth participant initially measured by a junior research assistant. Junior research assistants with more than two unreliable measurements were removed from the interviewing process and offered retraining.

For both studies, heads of households or guardians reported on demographics, health, and socioeconomic characteristics. To address self-reporting bias, survey instruments were carefully designed, piloted, and improved in the early stages of the study. Survey items were assessed for potential bias. Data on key variables were collected at the individual and household levels to address social desirability and selective recall bias. The main reasons for non-participation were hearing impairment, illness, nonresponse, refusal, and language barriers.

## Variables

Outcome variables included BAZ; HAZ; undernutrition, defined as BAZ < -2 standard deviations (SD) of the WHO 2007 growth reference; and stunting, defined as HAZ < -2SD of the WHO 2007 growth reference. Dichotomous variables categorizing individuals as undernourished (BAZ < -2SD) and stunted (HAZ < -2SD) were used for descriptive statistics. BAZ and HAZ were entered as continuous dependent variables in multivariate linear regression analyses.

Individual-level independent variables included age, sex, orphan status, school enrollment, work status, and health index. Age was calculated using the birth date and the date of survey administration. Sex was self-reported (females, 0; males, 1). Orphan status was determined based on death of either one parent or both parents and was verified at the household level (non-orphans, 0; orphans, 1). School enrollment was reported by the adolescent and verified at the household level (not enrolled, 0; enrolled, 1). Work status reflected participation in formal and informal work activities (not working, 0; working, 1).

Health index was based on self-reported susceptibility or incidence of malaria, persistent cough, and diarrhea. The rural data included self-reported incidence of diarrhea, cough, and malaria within the past 2–3 months, while the urban data reported on adolescents' perceived susceptibility to these illnesses. Discrepancies in health assessments across the two datasets are a limitation of this index. Health indicators were transformed into dichotomous variables for analysis, resulting in a health index score of 0 to 3 (no reported malaria, persistent cough, or diarrhea, 0; susceptibility or incidence of all three health conditions, 3).

Household level variables included place of residence, assets, and living conditions. Place of residence represented the primary predictor and was dichotomously categorized as urban (Moshi, 0) or rural (Kilosa, 1). Potential confounders included household assets and household living conditions, proxy indicators of wealth and socioeconomic status index. Household assets included ownership of a radio, bike, and/or motorbike, equally weighted and dichotomously classified. The asset index used a scale of 0–3 (poor/least affluent households, 0; most affluent households, 3). Household living conditions represented the sum of the following four indicators, each weighted equally and scored dichotomously: electricity, house flooring type (cement/stone was considered modern), source of drinking water (piped, public tap, and neighbor's water were considered clean water sources), and access to a sanitary toilet (flush, pour flush, and improved pit latrines were considered sanitary). These indicators are correlated with human welfare and economic development. A household with a fully modern or more affluent status had a living condition score of 4 with household access to electricity, modern flooring, clean water, and a sanitary toilet; a score of ≤ 1 indicates the poorest status with ≤ 1 of these resources.

## Statistical analysis

SPSS Version 26 (Armonk, NY, USA, 2019) was used to analyze the data. Median values, means, standard deviations, skewness, and kurtosis were estimated for continuous variables. Missing data met the assumptions for data missing at random within the respective datasets. Due to the complexity of collecting data across rural villages, missing data was higher in Kilosa/rural than in Moshi/urban. Missing data was minimized through return interviews conducted in Kilosa/rural. Multiple imputation was applied to height and weight, indicator variables for generating the dependent variables BAZ, HAZ, undernutrition, and stunting. To retain the sample size and reflect the characteristics of the merged dataset, missing data for dependent and independent variables were retained but were excluded in analyses.

Pearson's chi-squared test examined associations between undernutrition or stunting, health index, sociodemographic variables, and place of residence. An independent-samples t-test compared the means for BAZ and HAZ by rural/urban residence. Scatterplots, Pearson's correlation, and univariate linear regression analyses were applied to the data. Variables with p-values ≤ 0.25 were included in multivariate linear regression models.

It was assumed that there was no first order linear autocorrelation in the multiple linear regression data given Durbin-Watson statistics, and tolerance > 0.1 and VIF < 10 for all variables in the regression models. Furthermore, assessment of normality of the residuals with normal P-P plots indicated that residuals were normally distributed. We present the findings on the association between rural/urban residence by BAZ and HAZ, respectively, after adjusting for covariates. In multivariate linear regression analyses, Model 1, with BAZ as the outcome variable, adjusted for age, sex, household assets, and household living conditions. Model 2, with HAZ as the outcome variable, adjusted for age, sex, school enrolment, household assets, and household living conditions.

### Ethical considerations

Prior to participation, all adolescents provided their written assent and their parent(s) or guardian(s) provided written informed consent. The rural study was approved prior to any survey development or data collection by the Institutional Review Board at Tufts University and the Tanzania Commission for Science and Technology (COSTECH). Morogoro Regional Administration and Kilosa District Council granted field research approval. Research approval and ethical clearance for the urban study were obtained from Harvard University, the Ethical Clearance Committee of the Kilimanjaro Christian Medical College, and the Tanzanian National Institute of Medical Research. Ethics and data safety observations were provided by a Data Safety and Monitoring Board before the urban study began. Both studies were registered with COSTECH. This study was reviewed and approved by the University of Massachusetts at Amherst School of Public Health and Health Sciences Local Human Subjects Review Board before the study began.

## Results

This study included 1,125 adolescents aged 10–14 years (51.4% males, 48.6% females). Thirty-seven percent (36.7%) of adolescents lived in Kilosa/rural and 63.3% lived in Moshi/urban. Significant differences were found across rural and urban settings in orphan status (8.6% rural vs. 17.7% urban, $p < 0.001$), school enrollment (90% rural vs. 96.5% urban, $p < 0.001$), and work status (51% rural vs. 60.5% urban, $p = 0.002$) (Table 1).

Fifty-four percent (54.3%) of rural compared to 60.8% of urban adolescents were susceptible to or reported an incidence of malaria, persistent cough, or diarrhea ($p = 0.018$) (Table 1). Significant differences were observed for undernutrition (10.9% rural versus 5.1% urban, $p < 0.001$) and stunting (64.5% rural versus 3.1% urban, $p < 0.001$) (Table 1). Rural-urban differences in ownership of household assets and living conditions are reported in Table 2.

### Body mass index-for-age and sex z-scores (BAZ) and height-for-age and sex z-scores (HAZ)

The mean BAZ was lower among rural ($M$ = -0.98, $SD$ = 0.88) than urban ($M$ = -0.34, $SD$ = 1.03) adolescents; $t(1117)$ = 10.59, $p < 0.001$. The trajectory of median BAZ across age in urban females differed from their rural peers (data were smoothed), with the greatest rural disadvantage in BAZ observed at age 14 (Fig 1). Median BAZ of males was lower across age in rural and urban areas, with a sharper downward trajectory observed in rural males (Fig 1).

**Table 1. Characteristics of adolescents (10–14 yrs) in Kilosa/rural and Moshi/urban (*N* = 1,125).**

| Characteristics | $n^a$ | % of Total | Rural (%) | Urban (%) | $p^b$ |
|---|---|---|---|---|---|
| Overall | 1125 | 100 | 36.7 | 63.3 | |
| Age | | | | | 0.002 |
| 10 years | 298 | 26.6 | 24.5 | 27.8 | |
| 11 years | 220 | 19.6 | 16.9 | 21.2 | |
| 12 years | 256 | 22.8 | 21.5 | 23.6 | |
| 13 years | 217 | 19.3 | 25.7 | 15.7 | |
| 14 years | 131 | 11.7 | 11.4 | 11.8 | |
| Data missing | 3 | | 3 | 0 | |
| Sex | | | | | 0.920 |
| Male | 578 | 51.4 | 51.6 | 51.3 | |
| Female | 547 | 48.6 | 48.4 | 48.7 | |
| Orphan Status | | | | | <0.001 |
| Orphan | 161 | 14.4 | 8.6 | 17.7 | |
| Non-orphan | 955 | 85.6 | 91.4 | 82.3 | |
| Data missing | 9 | | 8 | 1 | |
| School Enrollment | | | | | <0.001 |
| Not enrolled | 66 | 5.9 | 10.0 | 3.5 | |
| Enrolled | 1054 | 94.1 | 90.0 | 96.5 | |
| Data missing | 5 | | 5 | 0 | |
| Work Status | | | | | 0.002 |
| Working | 637 | 57.0 | 51.0 | 60.5 | |
| Not working | 480 | 43.0 | 49.0 | 39.5 | |
| Data missing | 8 | | 7 | 1 | |
| **Health and Nutritional Indicators** | $n^a$ | % of Total | Rural (%) | Urban (%) | $p^b$ |
| Health Index$^c$ | | | | | 0.018 |
| No susceptibility or incidence of malaria, persistent cough, or diarrhea (0) | 463 | 41.6 | 45.7 | 39.2 | |
| One (1) of the following: malaria, persistent cough, or diarrhea | 313 | 28.1 | 29.6 | 27.2 | |
| Two (2) of the following: malaria, persistent cough, or diarrhea | 216 | 19.4 | 16.3 | 21.2 | |
| Three (3) of the following: malaria, persistent cough, or diarrhea | 122 | 11.0 | 8.4 | 12.4 | |
| Data missing | 11 | | 8 | 3 | |
| Undernutrition$^d$ | | | | | <0.001 |
| Not undernourished | 1038 | 92.8 | 89.1 | 94.9 | |
| Undernourished | 81 | 7.2 | 10.9 | 5.1 | |
| Data missing | 6 | | 1 | 5 | |
| Stunting$^e$ | | | | | <0.001 |
| Not stunted | 833 | 74.4 | 35.5 | 96.9 | |
| Stunted | 287 | 25.6 | 64.5 | 3.1 | |
| Data missing | 5 | | 2 | 3 | |

$^a$ Totals may differ due to missing data on some variables. Most missing data were in the rural setting, with the exception of undernutrition and stunting.

$^b$ Tests of statistical significance are based on two-tailed Pearson $\chi^2$, $p < 0.05$, $p < 0.01$, $p < 0.001$.

$^c$ Health Index defined as susceptibility to (urban) or incidence of (rural) diarrhea, malaria, or cough.

$^d$ Undernutrition defined as BMI-for-age $<$-2SD according to WHO 2007 reference.

$^e$ Stunting defined as height-for-age $<$-2SD according to WHO 2007 reference.

The mean HAZ was significantly lower among rural (*M* = -2.29, *SD* = 1.11) compared to urban (*M* = -0.09, *SD* = 1.17) adolescents; $t(1123) = 31.08$, $p < 0.001$. Median HAZ declined across age (data were smoothed) by sex and place of residence (Fig 2).

**Table 2. Household assets, housing conditions, water and sanitation in Kilosa/rural and Moshi/urban (N = 1,109).**

| Household Characteristics | $n^a$ | % of Total | Rural (%) | Urban (%) | $p^b$ |
|---|---|---|---|---|---|
| Overall | 1109 | 100 | 35.8 | 64.2 | |
| Assets | | | | | |
| Radio | 875 | 79.4 | 25.8 | 74.2 | <0.001 |
| Bicycle | 453 | 41.0 | 43.5 | 56.5 | <0.001 |
| Motorcycle | 261 | 23.6 | 1.9 | 98.1 | <0.001 |
| Housing Conditions | | | | | |
| Electricity | 417 | 37.2 | 9.1 | 90.9 | <0.001 |
| Modern Floor | 626 | 56.5 | 7.8 | 92.2 | <0.001 |
| Sanitary Toilets | 720 | 66.1 | 1.9 | 98.1 | <0.001 |
| Clean Water | 898 | 81.3 | 23.6 | 76.4 | <0.001 |

[a] Totals may differ due to missing data on some variables. Except for electricity, most missing data were in the rural area: Radio (23); bicycle (20); motorcycle (20); electricity (4); modern floor (18); sanitary toilet (35); clean water (20).

[b] Tests of statistical significance are based on two-tailed Pearson $\chi^2$, $p < 0.001$.

## Multivariate-adjusted BAZ and HAZ by rural/urban status

After adjusting for age, sex, household assets and living conditions, an inverse association was observed between BAZ and rural residence ($B$ = -0.29, 95% CI: -0.52, -0.07, $p$ = 0.01). The multiple regression model produced an $R^2$ = 0.13, $F(5,1045)$ = 32.19, $p < 0.001$ (Table 3). The regression equation can be illustrated as follows: a 12-year-old male living in a *poor rural* household would have an estimated BAZ of -1.18, or be mildly undernourished, while his peer living in a *poor urban* household would have a BAZ of -0.88, within the normal range of the WHO standard. If the 12-year-old lived in a modern rural household, his BAZ would be -0.60. For a 12-year-old female, the pattern was similar with an estimated BAZ of -0.95 in a poor rural household compared to -0.66 in an urban poor household and -0.37 in a rural modern household (Table 4).

An inverse association was observed between HAZ and rural residence, after adjusting for age, sex, school enrollment, household assets and living conditions ($B$ = -1.79, 95% CI: -2.03,

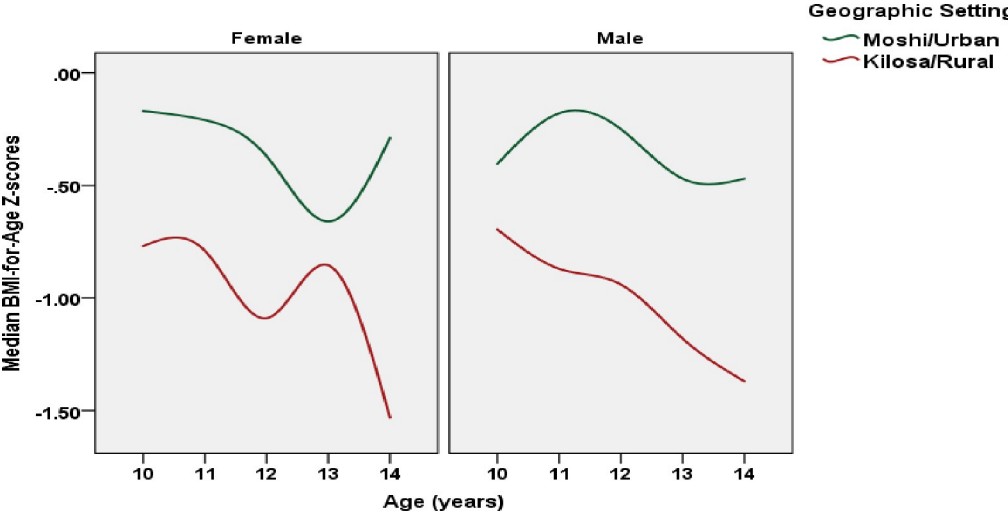

**Fig 1. Median BAZ for male and female adolescents (10–14 years old) living in Moshi and Kilosa (N = 1,119).**

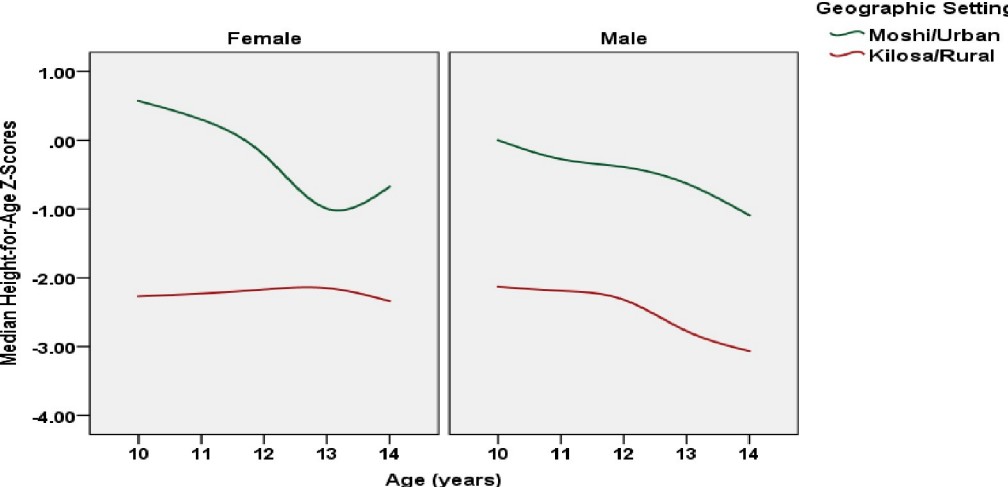

**Fig 2. Median HAZ for male and female adolescents (10–14 years old) living in Moshi and Kilosa (N = 1,125).**

-1.54, $p < 0.001$) (Table 3). The multiple regression model produced an $R^2 = 0.53$, $F(6,1046) =$ 198.79, $p < 0.001$. The regression equation is illustrated by the following example: A 12-year-old school-enrolled male living in a poor rural household would be stunted with an estimated HAZ of -2.54, while his school-enrolled peer living in a poor urban household would have a normal estimated HAZ of -0.75. Even if the school-enrolled 12-year-old male lived in a modern rural household, he would be stunted with an estimated HAZ of -1.91. The pattern is similar for a 12-year-old, school-enrolled female with an estimated HAZ of -2.32 in a poor rural household compared to -0.53 in a poor urban household or -1.69 in a modern rural household (Table 5).

## Discussion

The study aimed to investigate predictors of BAZ and HAZ, as well as the differences in magnitude of undernutrition and stunting, among 1,125 young adolescents living in rural and

**Table 3. Multivariate-adjusted BAZ and HAZ by rural/urban status among adolescents, 10–14 years, living in Tanzania (N = 1,125) [a].**

| Covariates | BAZ[b] N = 1050 | | | Covariates | HAZ[c] N = 1056 | | |
|---|---|---|---|---|---|---|---|
| | Coefficient (SE) | 95% CI | $p$[d] | | Coefficient (SE) | 95% CI | $p$[d] |
| (Intercept) | 0.231 (0.29) | -0.34, 0.80 | 0.428 | (Intercept) | 2.918 (0.36) | 2.21, 3.63 | <0.001 |
| Age | -0.084 (0.02) | -0.13, -0.04 | <0.001 | Age | -0.268 (0.02) | -0.32, -0.22 | <0.001 |
| Male | -0.226 (0.06) | -0.34, -0.11 | <0.001 | Male | -0.218 (0.07) | -0.35, -0.09 | 0.001 |
| | | | | School enrolled | -0.386 (0.16) | -0.69, -0.08 | 0.013 |
| Assets | 0.075 (0.03) | 0.01, 0.14 | 0.015 | Assets | 0.057 (0.03) | -0.01, 0.12 | 0.101 |
| Household living conditions index | 0.119 (0.04) | 0.04, 0.19 | 0.002 | Household living conditions index | 0.152 (0.04) | 0.07, 0.24 | <0.001 |
| Rural residence | -0.294 (0.11) | -0.52, -0.07 | 0.010 | Rural residence | -1.786 (0.13) | -2.03, -1.54 | <0.001 |

*Abbreviations*: BAZ body mass index-for age and sex z-score; *HAZ* height-for-age and sex z-score.

[a] Models using BAZ and HAZ, normalized using the two-step data transformation process, resulted in almost identical coefficients, SE, and p values to the original data.

[b] $R^2 = 0.13$; no change in $r^2$ was observed when rural residence was removed from this model.

[c] $R^2 = 0.53$; removal of rural residence from the model resulted in an $r^2 = 0.43$, suggesting that rural residence explained 10% of the variance in the model.

[d] Statistical significance assessed at the $p <0.05$, $p <0.01$, $p <0.001$ levels.

**Table 4. Estimated BAZ by rural/urban residence and household living conditions[a,b] for a reference adolescent.**

| Reference Adolescent | Rural or Urban | Living Conditions[a] | Estimated BAZ |
|---|---|---|---|
| 12-year-old male | Rural | Poor | -1.18 |
| | | Modern | -0.60 |
| | Urban | Poor | -0.884 |
| 12-year-old female | Rural | Poor | -0.95 |
| | | Modern | -0.37 |
| | Urban | Poor | -0.66 |

[a] Poor indicates presence ≤1 item in the household living condition index and no household assets. Modern indicates presence of all four items in the household living condition index and all three household assets.

urban settings of Tanzania. Rural adolescents experienced lower mean BAZ and HAZ, and were at higher risk of undernutrition and stunting, when compared to their urban peers. Place of residence, age, and sex were significant determinants of undernutrition, with rural males from households with fewer assets and poorer living conditions being most vulnerable to undernutrition compared to their female counterparts and urban peers. Similarly, there was a higher risk of stunting among rural adolescents, particularly among the youngest adolescents, males, and those living in households with few assets. These findings are consistent with the literature across Africa showing an urban advantage in BAZ [6–8] and HAZ [3,6–8] among adolescents.

The higher risk of undernutrition and stunting in rural adolescents may be related to poorer food and health access in rural settings [4,11]. Furthermore, access to and enrollment in school is better in urban areas [21], and both adolescent and maternal education offer a protective effect against poor nutritional status [2,4,5,22]. Malnutrition has adverse implications on health, chronic diseases, and economic productivity [1]. Better health is also positively associated with SES [2,4,13–15,17], and differences in SES across geographic location are thought to contribute to growing rural-urban health disparities [4,17]. However, these disparities persist after adjusting for sociodemographic factors [4,14,23], indicating that health is influenced by many contextual factors, and, as found in our study, place of residence is an independent and salient predictor of health outcomes. Furthermore, undernourished pre-adolescents may experience further deficits in reaching their height potential during adolescence, with differential implications for male and rural youth.

Household assets reported in our study showed similar trends to national data: 79% of our households owned a radio and 41% owned a bicycle, compared to 58% and 38%, respectively [24]. In general, urban households were more likely to own these assets than rural households.

**Table 5. Estimated HAZ by rural/urban status and household living conditions[a,b] for a reference school-enrolled adolescent.**

| Reference Adolescent | Rural or Urban | Living conditions[a] | Estimated HAZ |
|---|---|---|---|
| 12-year-old male enrolled in school | Rural | Poor | -2.536 |
| | | Modern | -1.909 |
| | Urban | Poor | -0.884 |
| 12-year-old female enrolled in school | Rural | Poor | -0.95 |
| | | Modern | -0.37 |
| | Urban | Poor | -0.66 |

[a] Poor indicates presence ≤1 item in the household living condition index and no household assets. Modern indicates presence of all four items in the household living condition index and all three household assets.

Improved hygiene and sanitation, which are indicative of SES and improved infrastructure [4,15,23] have been associated with better nutritional outcomes in previous studies [23], a finding corroborated by our study.

Food crises in SSA likely contribute to high prevalence rates of underweight, which was on the decline in Tanzania but is now increasing [25]. This raises public health concerns given that Tanzania has one of the largest rural-urban disparities in malnutrition [17], fueled by a falling rate of urban malnutrition despite rapid growth in cities [17] and limited progress on rural development. From 2004 to 2016, there was a slight increase in rural Tanzanian households with electricity (1% vs. 5%), sanitation facilities (1% vs. 14%), and access to improved water sources (22% vs. 25%) [24,26]. Rural and urban health is intricately linked, and impacted by accessibility to health care, improvements in infrastructure, and development policies.

This study responds to WHO and UNICEF requests for more research on younger adolescents, who are at a vulnerable stage in the life course due to sexual maturation, emerging health risks, and integration into adult lifestyles. Using a rigorous analytical framework, this study provides further evidence of an urban advantage in BAZ and HAZ among young adolescents in SSA, after controlling for household assets, household living conditions, and other covariates. Corroborated by previous studies [2,5,8,12], our results found that stunting was more pronounced in rural and urban males than in females.

The study design and large sample size are important strengths that add to the robustness and generalizability of this study's findings. The main limitation of this study is its cross-sectional design, which limits causal inference. Other limitations result from merging data from two separate studies in different regions of the country. The urban study utilized multilevel modeling to account for variation due to clustering of participants. Hence, the representativeness of samples varies due to different spatial diffusion of participants between the two studies. However, since adolescents in both studies were randomly selected, merging these datasets for comparative analysis was scientifically grounded and in alignment with previous epidemiological approaches.

Similarly, although the present study data was collected from two different regions in Tanzania, the disaggregated data on each district is informative and the comparison of districts is vital in providing country-level assessment. Furthermore, the comparison of nutritional status between Moshi and Kilosa seems appropriate based on the similar mean z-scores (±SD) for weight-for-height, height-for-age, and weight-for-age among children 0–59 months of age [27] in both Kilimanjaro (where Moshi is situated) and Morogoro (where Kilosa is situated) regions [28]. In lieu of a national reference for nutrition and health parameters, studies have recommended the use of health data from the Kilimanjaro region (and Moshi urban municipality) as a proxy for a national reference [29,30]. Using this framework, and by comparing the adolescents to the internationally-recognized WHO growth reference, we found that the actual difference in nutritional status between the two groups was striking and justifies our rural/urban comparison.

Another limitation of comparing data from two different regions may include the wide range of ethnic communities involved; Tanzania represents the highest level of ethnic diversity in SSA [31,32] with more than 120 ethnic or tribal groups. However, it is important to note that most of the population, including those in Kilosa and Moshi districts, is of Bantu heritage [33]. A high level of ethnic diversity and some overlap in tribal groups was reflected in both Kilosa and Moshi districts in the most recent national data that includes tribal or ethnic identity [34].

Maternal education, income, and food security status indicators, which have been associated with improved child nutritional outcomes and partly explain the urban health advantage [2,4,5,17], were unavailable in this study. These variables may have provided additional insight

on BAZ and HAZ differentials observed across rural/urban settings. Subjective self-reports of health status and sociodemographic indicators may result in upward bias in the estimated effect of these variables on undernutrition and stunting. Bias was minimized through rigorous processes in survey item development, question order, interviewer training, piloting and revising the survey, as well as querying both adolescents and their guardians. However, we were not able to fully utilize self-reported health data considering assessment methods differed across the datasets with the rural study employing a WHO clinical reference on self-reported health illnesses and the urban study assessing susceptibility to illnesses. Finally, assets varied with rural data reflecting a broader range of assets salient to this setting and urban data reflecting material assets. Since assets common to both geographic settings favored material assets, an urban bias may be reflected in this measurement.

## Conclusions

Undernutrition and stunting are global concerns that impact long-term health, quality of life, and productivity. The prevalence of undernutrition and stunting observed in this study, particularly in rural areas, raises concerns for adolescent health. The life course theoretical model (LCT), widely accepted in maternal and child health, posits that cumulative effects of biological, social, economic, and environmental risk factors are the underlying causes of persistent health inequities and poor health outcomes. However, conventional application of LCT is focused on the perinatal period and fails to connect the critical life stage of adolescence, an important stage of developmental processes that links childhood and young adulthood [35]. Nutrition research generally focuses on individual level factors and a comprehensive integration of social determinants of health, such as the place of residence, is needed to elucidate factors that may be addressed beyond the individual level [36].

Long-term improvements in the social determinants of malnutrition, such as SES, education, and rural infrastructure [2–5], can sustain gains in population health. Although child malnutrition is more likely to occur in poorer households in SSA [2,4,13–15,17,22], some analyses across SES levels have determined that intra-urban nutritional disparities are greater than intra-rural disparities [14,15]. Better understanding of social determinants of health, including identifying factors specific to place of residence and the extent to which these factors impact adolescent health, and situating these findings in the context of rural to urban migration, would be instrumental for developing policies to close the gap on rural-urban disparities.

Study findings indicate that both place of residence and an individual's status within that place are important determinants of nutritional status. Health interventions targeting rural populations, especially those in the lowest SES quintile, may have the greatest impact on improving the health of vulnerable adolescents. Finally, stunting-related obesity in urban areas should be carefully examined given Tanzania's high urban growth rate [10,17,37], high prevalence of rural stunting [11,15,21], and migration patterns [10,37]. National policies need to pay specific attention to the dual burden of disease stemming from the occurrence of undernutrition and overweight, and develop effective strategies for the prevention of chronic diseases [2]. Early adolescence provides critical opportunities to employ these strategies for health promotion and disease prevention.

## Supporting information

**S1 File. Moshi caregiver survey.**
(DOC)

**S2 File. Moshi adolescent survey.**
(DOC)

**S3 File. Kilosa adolescent survey.**
(DOC)

**S4 File. Kilosa caregiver survey.**
(DOC)

## Acknowledgments

The authors gratefully acknowledge Drs. Felton Earls and Mary Carlson for their guidance and contribution to this manuscript, including the use of the urban adolescent data and, more importantly, for their commitment to the health and well-being of children around the world. The authors would like to thank the research teams, community members, and participants in Tanzania, as well as the Tanzania Food and Nutrition Center and District offices for their support and COSTECH for study approval.

## Author Contributions

**Conceptualization:** Lorraine S. Cordeiro, Nicholas P. Otis, Lindiwe Sibeko, Jerusha Nelson-Peterman.

**Data curation:** Lorraine S. Cordeiro.

**Formal analysis:** Lorraine S. Cordeiro, Nicholas P. Otis.

**Funding acquisition:** Lorraine S. Cordeiro, Nicholas P. Otis.

**Investigation:** Lorraine S. Cordeiro, Nicholas P. Otis, Lindiwe Sibeko.

**Methodology:** Lorraine S. Cordeiro, Nicholas P. Otis, Lindiwe Sibeko, Jerusha Nelson-Peterman.

**Project administration:** Lorraine S. Cordeiro, Jerusha Nelson-Peterman.

**Resources:** Lorraine S. Cordeiro.

**Software:** Lorraine S. Cordeiro.

**Supervision:** Lorraine S. Cordeiro.

**Validation:** Lorraine S. Cordeiro, Nicholas P. Otis, Jerusha Nelson-Peterman.

**Visualization:** Lorraine S. Cordeiro, Nicholas P. Otis, Lindiwe Sibeko, Jerusha Nelson-Peterman.

**Writing – original draft:** Lorraine S. Cordeiro, Nicholas P. Otis, Lindiwe Sibeko.

**Writing – review & editing:** Lorraine S. Cordeiro, Nicholas P. Otis, Lindiwe Sibeko, Jerusha Nelson-Peterman.

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
