## [Decision Letter · Decision Letter 0]

16 Aug 2021

PONE-D-21-20245

Rural-urban disparities in the nutritional status of younger adolescents in Tanzania

PLOS ONE

Dear Dr. Cordeiro,

Thank you for submitting your manuscript to PLOS ONE. After careful consideration, we feel that it has merit but does not fully meet PLOS ONE’s publication criteria as it currently stands. Therefore, we invite you to submit a revised version of the manuscript that addresses the points raised during the review process.

ACADEMIC EDITOR: Considering my own reading of the paper and reviewers opinion, I am in favour of recommending this paper subject to the revisions as suggested by reviewer-2.

We look forward to receiving your revised manuscript.

Kind regards,

Srinivas Goli, Ph.D.

Academic Editor

PLOS ONE

Journal Requirements:

a) Did participants provide their written or verbal informed consent to participate in this study?

3. Please include additional information regarding the survey or questionnaire used in the study and ensure that you have provided sufficient details that others could replicate the analyses. For instance, if you developed a questionnaire as part of this study and it is not under a copyright more restrictive than CC-BY, please include a copy, in both the original language and English, as Supporting Information. If the original language is written in non-Latin characters, for example Amharic, Chinese, or Korean, please use a file format that ensures these characters are visible.

4. Please state whether you validated the questionnaire prior to testing on study participants. Please provide details regarding the validation group within the methods section.

Additional Editor Comments:

Considering my own reading of the paper and reviewers opinion, I am in favour of recommending this paper subject to the revisions as suggested by reviewer-2.

Reviewers' comments:

Reviewer's Responses to Questions

**Comments to the Author**

1. Is the manuscript technically sound, and do the data support the conclusions?

Reviewer #1: Partly

Reviewer #2: Yes

2. Has the statistical analysis been performed appropriately and rigorously? 

Reviewer #1: Yes

Reviewer #2: Yes

3. Have the authors made all data underlying the findings in their manuscript fully available?

Reviewer #1: Yes

Reviewer #2: Yes

4. Is the manuscript presented in an intelligible fashion and written in standard English?

Reviewer #1: No

Reviewer #2: Yes

5. Review Comments to the Author

Reviewer #1: This study has touched an important and interesting topic on adolescents. But why it has considered the age group of 10-14 years only, while, the WHO reports mentioned 10-19 years. Is there any particular reason to take this age group only?

Reviewer #2: 1. While the researcher argue that there is lack of information on urba-rural adolescents nutrition, Their literature review shows that studies of that type were conducted in Sub-saharan Africa. This includes studies in Nigeria, Cameroon and Ethiopia. This create contradiction which calls for more thorough review to ensure that the gap is clearly seen.

2. I feel that this study has applied very inapropriate comparison. It could have been importat to compare urban rural within one region at least to capture advantages of one area against another area sharing similar characteristics but diffferent in geographical setting. Kilosa rural is too far from Moshi and has people of different cultures and other developmental milestones Basing on this grount, the differences between Kilosa and Moshi are too obvious and does not warrrant any study. It could have been prroper if Moshi urban was compared against Moshi rural where people are similar but has differences in Geography. Or Comparison could have been executed between Kilosa rural and Kilosa urban to see the differences. Further another way could have been comparing Kilosa rural and Morogoro which is the capital of the same region.

3. Kilosa and Moshi are in areas of too different diesease profile. Whereas Kilosa is in area of high prevance of communicable disease, Moshi is not. For instance,prevalance of Malaria in Moshi is very low as compared to high prevance in Kilosa. This has serious implication in nutrition and has been documented in other national reports. It could be good if the comparison was conducted in areas where there is commonalities. the health parameters used for comparison are innapropriate and cannot provide credible results

4. School enrollment and other developmental milestones between Kilosa and even rural areas of Moshi are too different from each other. One should expect more differences between Kilosa rural and Moshi urban. I feel that some of the parameters used for comparison have yielded very obvios results

5.The differences between Kilosa and Moshi are too obvious and therefore, if one is to make comparison, tha analysis should focus on parameters which are not obvious. Otherwise, the comparison should be within same zone if the need is to compare geographical areas. It is crucial for researchers to think on how best to present this comparison avoiding geographical comparisons in areas which are not sharing common characteristics

6. PLOS authors have the option to publish the peer review history of their article (what does this mean?). If published, this will include your full peer review and any attached files.

Reviewer #1: No

Reviewer #2: No

---

## [Author Response · Author response to Decision Letter 0]

9 Nov 2021

Also included in word format under 'attach files'.

September 27, 2021

Dear Dr. Goli,

We found the comments of the reviewers to be thorough and helpful in improving our manuscript. We have carefully considered and addressed each comment as outlined below. We thank you and the reviewers for the comments which have significantly improved the manuscript.

We have also included out Financial Disclosure statement below:

This study was funded with the support of UNICEF/ Tanzania; U.S. National Institute of Mental Health (R01 MH66801), and the University of Massachusetts Amherst. The funders had no role in study design, data collection and analysis, decision to publish, or preparation of the manuscript.

Best,

Lorraine S. Cordeiro and Nicholas P. Otis

Response to Editorial Comments

1. Formatting has been addressed.

2. Ethics: Inserted statement about written informed assent and consent in the Ethics Section of submission and in manuscript.

3. Please include additional information regarding the survey or questionnaire used in the study and ensure that you have provided sufficient details that others could replicate the analyses. For instance, if you developed a questionnaire as part of this study and it is not under a copyright more restrictive than CC-BY, please include a copy, in both the original language and English, as Supporting Information. If the original language is written in non-Latin characters, for example Amharic, Chinese, or Korean, please use a file format that ensures these characters are visible.

Inserted: Adolescents and their caregivers were interviewed at the adolescent's residence or schools. Surveys in both English and Swahili are included as supporting information.

Supporting Information: S1 File: Kilosa survey in English and S2 File: Moshi survey in English.

Please state whether you validated the questionnaire prior to testing on study participants. Please provide details regarding the validation group within the methods section.

Inserted: The questionnaires were pre-tested among a sample of non-participants in Kilosa District and Moshi prior to using it with study participants. Trained interviewers administered the structured, pre-tested questionnaires in Swahili. The survey tool was developed in English based on an extensive literature review, and translated into Swahili by language experts in Tanzania, and then back-translated into English. We conducted content validity, ensuring that terms and meanings were understood by participants in the pretest and that terms were accurately translated from English to Swahili. Construct validity required setting a priori hypotheses of associations and analyzing the pilot data to see if these hypotheses signaled validity. For quality assurance, piloting the survey tool among adolescents and their parents/guardian in both Kilosa and Moshi resulted in no major changes in the content and construct of the survey.

4. We note that the grant information you provided in the ‘Funding Information’ and ‘Financial Disclosure’ sections do not match. When you resubmit, please ensure that you provide the correct grant numbers for the awards you received for your study in the ‘Funding Information’ section.

Authors Response: Corrected funding information and financial disclosure so they matched. 

This statement is required for submission and will appear in the published article if the submission is accepted. Please make sure it is accurate and that any funding sources listed in your Funding Information later in the submission form are also declared in your Financial Disclosure statement. 

Authors Response: The authors have declared that no competing interests exist.

Describe where the data may be found in full sentences. If you are copying our sample text, replace any instances of XXX with the appropriate details. 

Authors Response: All urban data files and research materials are available from the Harvard University Data Repository at https://dataverse.harvard.edu/. Access to rural data can be requested directly from the corresponding author and surveys are included in this manuscript under Supporting Information.

Authors Response: The reference list is complete and correct.

Response to Reviewers

1. Is the manuscript technically sound, and do the data support the conclusions?

Reviewer #1: Partly

Reviewer #2: Yes

Authors response: No response is needed for this question.

2. Has the statistical analysis been performed appropriately and rigorously? 

Reviewer #1: Yes

Reviewer #2: Yes

Authors response: No response is needed for this question.

3. Have the authors made all data underlying the findings in their manuscript fully available?

Reviewer #1: Yes

Reviewer #2: Yes

Authors response: All urban data files and research materials are available from the Harvard University Data Repository at https://dataverse.harvard.edu/. Access to rural data can be requested directly from the corresponding author and surveys are included in this manuscript under Supporting Information.

4. Is the manuscript presented in an intelligible fashion and written in standard English?

Reviewer #1: No

Reviewer #2: Yes

Authors response: No response is needed for this question.

5. Review Comments to the Author

Please use the space provided to explain your answers to the questions above. You may also include additional comments for the author, including concerns about dual publication, research ethics, or publication ethics. (Please upload your review as an attachment if it exceeds 20,000 characters).

Reviewer #1: This study has touched an important and interesting topic on adolescents. But why it has considered the age group of 10-14 years only, while, the WHO reports mentioned 10-19 years. Is there any particular reason to take this age group only?

Authors response: We appreciate this comment from Reviewer #1. The urban dataset only included adolescents aged 10-14 years, while the rural dataset included individuals aged 10-19 years. For comparative purposes, this study examined data on adolescents aged 10-14 years which was available in both datasets. We justify this focus on younger adolescents based on the WHO recommendation for more research on this age group (i.e. <15 years of age). WHO states that “there is a pressing need for research in this area in low and middle income countries to understand about the realities of young adolescents lives, especially in the context of the rapid changes that are occurring in societies.”1 Furthermore, the Guttmacher Institute identified key research gaps in health information noting that “excluded groups of adolescents in developing regions include adolescents younger than 15, unmarried/never-married women, youth in vulnerable situations, male adolescents...”2 Data on 10-19 year old participants in the rural dataset are published elsewhere.3

1. https://www.who.int/reproductivehealth/topics/adolescence/very_young_ados/en/

2. https://www.guttmacher.org/report/research-gaps-in-sexual-and-reproductive-health

3. Cordeiro LS, Wilde PE, Semu H, Levinson FJ. Household food security Is inversely associated with undernutrition among adolescents from Kilosa, Tanzania. Journal of Nutrition, 2012;142: 1741–1747. doi:10.3945/jn.111.155994

Reviewer #2: 

1. While the researcher argue that there is lack of information on urba-rural adolescents nutrition, Their literature review shows that studies of that type were conducted in Sub-saharan Africa. This includes studies in Nigeria, Cameroon and Ethiopia. This create contradiction which calls for more thorough review to ensure that the gap is clearly seen.

Authors response: Reviewer#2 raised a valid and important issue. We conducted a comprehensive review of the literature based on the comments from Reviewer #2. We concluded that a robust literature base is still in its infancy in regards to adolescent nutrition in Sub Saharan Africa (and truly adolescents in general); however, literature on this population subgroup is steadily growing. There are relatively few adolescent studies on urban-rural comparisons in this region of the world, and most are either in West Africa or in Ethiopia. Two studies in Ethiopia, one in Nigeria, and one in Cameroon are presented in this manuscript, indicating limited studies available to draw conclusions about urban-rural disparities in adolescent nutritional status across Sub Saharan Africa. This study presents new data from Tanzania, a low-income East African country with a political economy and demographical profile that is different from Ethiopia and West African nations, and adds to the literature base on adolescent nutrition and urban-rural disparities in Sub Saharan Africa.

2. I feel that this study has applied very inapropriate comparison. It could have been importat to compare urban rural within one region at least to capture advantages of one area against another area sharing similar characteristics but diffferent in geographical setting. Kilosa rural is too far from Moshi and has people of different cultures and other developmental milestones Basing on this grount, the differences between Kilosa and Moshi are too obvious and does not warrrant any study. It could have been prroper if Moshi urban was compared against Moshi rural where people are similar but has differences in Geography. Or Comparison could have been executed between Kilosa rural and Kilosa urban to see the dfferences. Further another way could have been comparing Kilosa rural and Morogoro which is the capital of the same region.

Authors response: We agree with Reviewer # 2 that comparing adolescents in the same region of the country would have been optimal. Reviewer #2 raises an issue that allows us to strengthen our paper and we appreciate the thoughtful comments. To address the concerns raised by Reviewer #2, we present our rationale for the comparison between Kilosa and Moshi below. 

a. Tanzania represents the highest level of ethnic diversity in SSA1,2 with 120+ ethnic or tribal groups, however, it important to note that most of the population is of Bantu heritage.3 So, while Kilosa District is in a different region than Moshi, it comprises a large population of the dominant ethnic groups found in Moshi District and a vast majority of the population in both districts is of Bantu heritage. There were 54 ethnicities represented in the Kilosa rural dataset and the Moshi urban data also had a high level of ethnic diversity. We also decided to examine national data more closely and requested access to DHS data. In an analysis of the 1996 DHS dataset representing the most recent national DHS survey that includes tribal or ethnic identity, there were over 25 different tribes in both Kilimanjaro and Morogoro regions with overlap in several groups across the regions. For context, Miguel (2004) notes that Tanzanian leadership focused on national unity across ethnic identities soon after independence4, leading to insignificant effects in political representation and financial distribution that could be caused by politicization of ethnicity.1 We also recognize that there is a lack of country data disaggregating health outcomes by ethnicity due to the Tanzanian socialist ideology that has successfully emphasized national identity over ethnicity.5 

1.Weber, A. (2010). The causes of politicization of ethnicity: A comparative case study of Kenya and Tanzania. APSA 2010 Annual Meeting Paper. doi:10.5167/uzh-63126

2.Fearon, J. D. (2003) Ethnic and Cultural Diversity by Country. Economic Growth, 8, 195–222. doi:10.1023/A:1024419522867

3.Goldberg 2020 Country-Report-Tanzania_Goldberg.pdf. (n.d.). Retrieved August 8, 2021, from https://rad-aid.org/wp-content/uploads/Country-Report-Tanzania_Goldberg.pdf

4.Miguel, E. (2004). Tribe or nation? Nation building and public goods in Kenya versus Tanzania. World Politics, 56(3), 327-362. doi:10.1017/S0043887100004330

5.Lawson, D. W., Borgerhoff Mulder, M., Ghiselli, M. E., Ngadaya, E., Ngowi, B., Mfinanga, S. G., ... & James, S. (2014). Ethnicity and child health in northern Tanzania: Maasai pastoralists are disadvantaged compared to neighbouring ethnic groups. PloS one, 9(10), e110447. doi:10.1371/journal.pone.0110447

b. The comparison of the rural and urban data presents one focus on this manuscript and the main focus is the findings from each area of Tanzania. In lieu of a national reference for nutrition and health parameters, studies have presented health data from the Kilimanjaro region (and Moshi Urban Municipality) and recommend the use of this data as a proxy for a national reference. For example, in a study on infants, children, and adolescents in Moshi, Kilimanjaro region, Buchanan et al. (2010) reported that “data regarding immunological and haematological reference intervals for healthy African populations are scarce, particularly for infants, children, and adolescents. Values currently used are often based upon results generated from Caucasian populations living in industrialized countries (Wintrobe 1981; Tugume et al. 1995; Karita et al. 2009).”1 Their study aimed to “either verify or establish normal reference ranges, for haematological and immunological indices among healthy Tanzanian children…[and their study provided] further evidence that the establishment of local reference ranges is critical for optimal patient management and medical research. Reference intervals derived primarily from Caucasians residing in developed nations, in particular, are inappropriate for this population.”1 They also provide biochemistry reference values using data for healthy children and adolescents in the Kilimanjaro region of Tanzania.2 Using this framework and by comparing the Tanzanian rural and urban adolescents to the internationally-recognized WHO growth reference, we note that the actual difference in nutritional status between the two groups is striking and justifies that this comparison presents valuable information on disparities in Tanzania. 

1.Buchanan, A. M., Muro, F. J., Gratz, J., Crump, J. A., Musyoka, A. M., Sichangi, M. W., ... & Cunningham, C. K. (2010). Establishment of haematological and immunological reference values for healthy Tanzanian children in Kilimanjaro Region. Tropical Medicine & International Health, 15(9), 1011-1021. doi:10.1111/j.1365-3156.2010.02585.x

2.Buchanan, A. M., Fiorillo, S. P., Omondi, M. W., Cunningham, C. K., & Crump, J. A. (2015). Establishment of biochemistry reference values for healthy Tanzanian infants, children and adolescents in Kilimanjaro Region. Tropical Medicine & International Health, 20(11), 1569-1577. doi:10.1111/tmi.12580

3. Kilosa and Moshi are in areas of too different diesease profile. Whereas Kilosa is in area of high prevance of communicable disease, Moshi is not. For instance, prevalance of Malaria in Moshi is very low as compared to high prevance in Kilosa. This has serious implication in nutrition and has been documented in other national reports. It could be good if the comparison was conducted in areas where there is commonalities. the health parameters used for comparison are innapropriate and cannot provide credible results.

Authors response: 

a. We addressed the concerns regarding comparison of the two districts above.

b. Since self-reported health is generally a poor index of health and not necessarily a salient predictor of undernutrition in this area, we recommend that the health parameters based on self-reported data be viewed in this context and have discussed this further in our limitations. 

c. In 2018, 94% of global malaria deaths occurred in SSA1. In 2018, case fatality of malaria contributed to 18% of under five mortality in SSA1 and malaria likely has a negligible effect on undernutrition in adolescence. Research on the malaria-malnutrition interactions indicate a complex relationship with inconclusive findings. Some studies report no association between malaria and malnutrition2 while others find an association. Das et al. (2015) concluded from a systematic review of the literature that “the evidence on the effect of malnutrition on malaria risk remains inconclusive… Further clarification on malaria-malnutrition interactions would also serve as a basis for designing [interventions].”3

1.Ouédraogo, M., Kangoye, D. T., Samadoulougou, S., Rouamba, T., Donnen, P., & Kirakoya-Samadoulougou, F. (2020). Malaria case fatality rate among children under five in Burkina Faso: an assessment of the spatiotemporal trends following the implementation of control programs. International journal of environmental research and public health, 17(6), 1840. doi:10.3390/ijerph17061840

2.Charchuk, R., Houston, S., & Hawkes, M. T. (2015). Elevated prevalence of malnutrition and malaria among school-aged children and adolescents in war-ravaged South Sudan. Pathogens and global health, 109(8), 395-400. doi:10.1080/20477724.2015.1126033

3.Das, D., Grais, R. F., Okiro, E. A., Stepniewska, K., Mansoor, R., Van Der Kam, S., ... & Guerin, P. J. (2018). Complex interactions between malaria and malnutrition: a systematic literature review. BMC medicine, 16(1), 1-14. doi:10.1186/s12916-018-1177-5

d. We note that the comparison nutritional status between Moshi and Kilosa seems appropriate based on the similar mean z-scores (±SD) for weight-for-height, height-for-age, and weight-for-age among children 0-59 months of age1 in both Kilimanjaro (where Moshi is situated) and Morogoro (where Kilosa is situated) regions.2 

 Kilimanjaro Morogoro

Weight for height 0.05 ± 0.99 -0.06 ± 1.09

Height for age -1.09 ± 1.12 -1.26 ±1.13

Weight for age -0.58 ± 1.05 -0.74 ± 1.09

Source: Tanzania National Nutrition Survey 20182 

1.https://www.who.int/tools/growth-reference-data-for-5to19-years 2.https://www.unicef.org/tanzania/reports/tanzania-national-nutrition-survey-2018

4. School enrollment and other developmental milestones between Kilosa and even rural areas of Moshi are too different from each other. One should expect more differences between Kilosa rural and Moshi urban. I feel that some of the parameters used for comparison have yielded very obvios results.

Authors response: We appreciate this relevant comment from Reviewer #2 and have presented the rationale for the comparison between Kilosa and Moshi in our response to comments #2 and #3 above.

5.The differences between Kilosa and Moshi are too obvious and therefore, if one is to make comparison, tha analysis should focus on parameters which are not obvious. Otherwise, the comparison should be within same zone if the need is to compare geographical areas. It is crucial for researchers to think on how best to present this comparison avoiding geographical comparisons in areas which are not sharing common characteristics.

Authors response: We appreciate this relevant comment from Reviewer #2 and have presented the rationale for the comparison between Kilosa and Moshi in our response to comments #2 and #3 above.

6. PLOS authors have the option to publish the peer review history of their article (what does this mean?). If published, this will include your full peer review and any attached files.

Do you want your identity to be public for this peer review? For information about this choice, including consent withdrawal, please see our Privacy Policy.

Reviewer #1: No

Reviewer #2: No

Authors response: No response is needed for this question.

Authors response: All figures were uploaded to PACE and included in this revision.

---

## [Decision Letter · Decision Letter 1]

3 Dec 2021

Rural-urban disparities in the nutritional status of younger adolescents in Tanzania

PONE-D-21-20245R1

Dear Dr. Cordeiro,

We’re pleased to inform you that your manuscript has been judged scientifically suitable for publication and will be formally accepted for publication once it meets all outstanding technical requirements.

Kind regards,

Srinivas Goli, Ph.D.

Academic Editor

PLOS ONE

Additional Editor Comments (optional):

Considering the reviewers suggestion and my own reading of this paper, I am recommending it for publication in PLOS One.

Reviewers' comments:

Reviewer's Responses to Questions

**Comments to the Author**

1. If the authors have adequately addressed your comments raised in a previous round of review and you feel that this manuscript is now acceptable for publication, you may indicate that here to bypass the “Comments to the Author” section, enter your conflict of interest statement in the “Confidential to Editor” section, and submit your "Accept" recommendation.

Reviewer #1: All comments have been addressed

Reviewer #2: All comments have been addressed

2. Is the manuscript technically sound, and do the data support the conclusions?

Reviewer #1: Yes

Reviewer #2: Yes

3. Has the statistical analysis been performed appropriately and rigorously? 

Reviewer #1: Yes

Reviewer #2: Yes

4. Have the authors made all data underlying the findings in their manuscript fully available?

Reviewer #1: Yes

Reviewer #2: Yes

5. Is the manuscript presented in an intelligible fashion and written in standard English?

Reviewer #1: Yes

Reviewer #2: Yes

6. Review Comments to the Author

Reviewer #1: (No Response)

Reviewer #2: The authors have adressed most of the comments in a satisfactory manner . This paper may be ready for publication

7. PLOS authors have the option to publish the peer review history of their article (what does this mean?). If published, this will include your full peer review and any attached files.

Reviewer #1: No

Reviewer #2: No

---

## [Editor Report · Acceptance letter]

9 Dec 2021

PONE-D-21-20245R1 

Rural-urban disparities in the nutritional status of younger adolescents in Tanzania 

Dear Dr. Cordeiro:

I'm pleased to inform you that your manuscript has been deemed suitable for publication in PLOS ONE. Congratulations! Your manuscript is now with our production department. 

Kind regards, 

on behalf of

Dr. Srinivas Goli 

Academic Editor

PLOS ONE